# Removal of Chromium(III) and Cadmium(II) Heavy Metal Ions from Aqueous Solutions Using Treated Date Seeds: An Eco-Friendly Method

**DOI:** 10.3390/molecules26123718

**Published:** 2021-06-18

**Authors:** Mohammad Azam, Saikh Mohammad Wabaidur, Mohammad Rizwan Khan, Saud I. Al-Resayes, Mohammad Shahidul Islam

**Affiliations:** Department of Chemistry, College of Science, King Saud University, P.O. Box 2455, Riyadh 11451, Saudi Arabia; swabaidur@ksu.edu.sa (S.M.W.); mrkhan@ksu.edu.sa (M.R.K.); sresayes@ksu.edu.sa (S.I.A.-R.); mislam@ksu.edu.sa (M.S.I.)

**Keywords:** date pits, adsorption, metal ions, model studies

## Abstract

The aim of the research was to prepare low-cost adsorbents, including raw date pits and chemically treated date pits, and to apply these materials to investigate the adsorption behavior of Cr(III) and Cd(II) ions from wastewater. The prepared materials were characterized using SEM, FT-IR and BET surface analysis techniques for investigating the surface morphology, particle size, pore size and surface functionalities of the materials. A series of adsorption processes was conducted in a batch system and optimized by investigating various parameters such as solution pH, contact time, initial metal concentrations and adsorbent dosage. The optimum pH for achieving maximum adsorption capacity was found to be approximately 7.8. The determination of metal ions was conducted using atomic adsorption spectrometry. The experimental results were fitted using isotherm Langmuir and Freundlich equations, and maximum monolayer adsorption capacities for Cr(III) and Cd(II) at 323 K were 1428.5 and 1302.0 mg/g (treated majdool date pits adsorbent) and 1228.5 and 1182.0 mg/g (treated sagai date pits adsorbent), respectively. It was found that the adsorption capacity of H_2_O_2_-treated date pits was higher than that of untreated DP. Recovery studies showed maximal metal elution with 0.1 M HCl for all the adsorbents. An 83.3–88.2% and 81.8–86.8% drop in Cr(III) and Cd(II) adsorption, respectively, were found after the five regeneration cycles. The results showed that the Langmuir model gave slightly better results than the Freundlich model for the untreated and treated date pits. Hence, the results demonstrated that the prepared materials could be a low-cost and eco-friendly choice for the remediation of Cr(III) and Cd(II) contaminants from an aqueous solution.

## 1. Introduction

Over the last decade, the identification of toxic elements, particularly in water, has grown into a key issue of public interest [1]. Among them, heavy metals have received great attention because of their deadly nature [2]. Heavy metals are increasingly worrisome due to their prospective influences on the environment and human health [2,3,4]. Owing to their physical and chemical properties, they have are extensively disseminated into the surroundings and are used in industrial areas including agricultural, medical and domestic [5,6,7]. Industrial effluents are a major source of environmental pollution and exposure to human beings [8]. Heavy metals are frequently present in water bodies, and most of them are regarded as highly lethal even in very small amounts [9,10]. Heavy metals, for instance, lead, cadmium, chromium, cobalt, arsenic, mercury and nickel, are listed as very toxic elements [11]. These metals are assumed to be ubiquitous toxicants, and their deadliness is influenced by many factors, for instance, the amounts of heavy metal doses, types of elements and exposure route [12,13,14]. Heavy metals can be transported into the human body via many sources, for instance, water, air, skin and food, and can become more lethal in the human body when they are not completely digested and stored in the muscle [15,16]. They may damage various vital human organs such as gastrointestinal, kidneys, bones, endocrine glands and central nervous system [17,18,19,20]. Prolonged exposure to heavy metal has been associated with a number of deteriorating illnesses of these human vital organs and may increase the threat of cancer diseases [19,21]. A few heavy metals are also important for life and show a unique role in the metabolic system of human beings, for instance, the functioning of of acute enzyme locations; however, they can damage the organism in higher amounts [22]. Based on their carcinogenicity in experimental animals, some heavy metals (cadmium, arsenic, nickel and chromium) have been categorized as probable human carcinogens (Group 1) by the International Agency for Research on Cancer (IARC) and the Environmental Protection Agency (EPA) [23].

To standardize the uncontrollable release of these unsafe toxins in water, either originating from natural or wastewater sources, innovative and modern water treatment tools are offered worldwide. Numerous methods, for instance, ion exchange, membrane filtration, sola water evaporation, electrochemical process and others, have been applied to eradicate these lethal chemicals [24,25,26,27]. Nevertheless, the conventional methods seem to be inappropriate to apply in the elimination of heavy metals at low levels. In addition, they harm both natural life and the environment. In recent years, the adsorption method has been found to be appropriate for the removal of contaminants from polluted water samples. Moreover, the adsorption method was found to be low-cost, easy to use, eco-friendly, and highly effective compared to earlier methods [28,29,30]. The adsorption method comprises the separation and aggregation of target analytes from one stage to another [30]. The adsorption system is quite easy to work with and is very effective in the elimination of deadly contaminants at low levels [30,31].

In the present work, we prepared low-cost adsorbents from date pits and applied them for the removal of potential heavy metals (cadmium and chromium) from aqueous solutions. Date seeds are abundant in Saudi Arabia, and the presence of various functional groups such as carboxylic acid, ester, carboxylate, hydroxyl, phenolic and amino in the materials allow them to serve as potential adsorbents for the removal of heavy metal ions. The negative (–OH) surface functionalities of the as-prepared date pit powders were enhanced by treating them with hydrogen peroxide overnight with a magnetic stirrer. SEM, BET and FT-IR techniques were used to analyze the as-prepared and treated samples, confirming the excellent surface morphology and particle size of the adsorbent materials. The H_2_O_2_-treated date pits adsorbents showed outstanding adsorption capabilities, suggesting that the materials could be effective for the removal of toxic metals (Cr(III) and Cd(II)) from aqueous solutions.

## 2. Materials and Methods

### 2.1. Chemicals and Reagents

Experiments were carried out using a digital weight balance, 1000 mL beaker, 100 mL conical flask, funnel and 1000 mL volumetric flask. Analytical-grade solvents and chemicals were used throughout the experiments. The chloride salts of metals were procured from BDH chemical supplier (Poole, UK). Sodium hydroxide (NaOH) and hydrochloric acid, which were used for the pH adjustment, were provided by Sigma-Aldrich, St. Louis, MO, USA. Sulfuric acid (H_2_SO_4_), nitric acid (HNO_3_) and ammonium hydroxide (NH_4_OH) were bought from Merck, Germany. Hydrogen peroxide (H_2_O_2_) solution of 30% *v*/*v* was purchased Sigma-Aldrich, Taufkirchen, Germany. The DI water used during the whole experiment was produced from milli-Q water purification system (USA). Unless otherwise stated, all other chemicals used in the experiments were analytical grade.

### 2.2. Instrumentation

The point of zero charge was determined using the salt-addition method (pHPZC). SEM (SEM, Jeol JSM 5400 LV, Jeol Ltd., Tokyo, Japan) was used to examine the surface morphology of the as-prepared date pits and the treated date pits materials. The surface was smooth. Fourier transform infrared spectroscopy (FT-IR, Thermo Scientific Nicolet 6700) with KBr dilution at 1:100 weight ratio corresponding to the wave numbers 400 and 4000 cm^−1^ were recorded with an average of 32 scans with a resolution of ±4 cm^−1^ to explore the surface functionalities of the groups present over the synthesized adsorbents. The pore size and surface area of samples were determined using a BET surface area analyzer (Micromeritics–Gemini VII 2390 V1.03). A multifunctional X-ray diffractometer was used to determine the surface crystallinity and average particle size of the samples (XRD, Ultima IV, Rigaku).

### 2.3. Preparation of Standard Solution

The volumetric flask (1000 mL) was cleaned, oven-dried and labeled. All the stock standard solutions of metal ions including Cr(III) and Cd(II) of concentration 1000 ppm were prepared individually in DI water by accurately weighing their respective salts. The stock solutions were then diluted to the desired concentration ranges using the continuous dilution method. For calibration purposes, the sample was diluted to 10 ppm, 6 ppm, 4 ppm and 2 ppm.

### 2.4. Preparation of Biosorbent

Two types of dates (7 of each kg) were purchased from different local market stores in Riyadh, Saudi Arabia. The varieties of dates chosen were majdool and sagai. The seeds were separated from their original date fruits. Pictures of the collected date samples and seeds are shown in Figure 1. The separated seeds were rinsed several times with clean tap water and then with distilled water (DI) to eliminate any traces of consumable parts, dirt and dust from them. Then, the clean seeds were dried in sunlight for several days and then oven-dried at 80 °C for 2 h. Then, the dried seeds were taken out from the oven and mechanically crushed using hammering and blending (Moulinex, 700 W) to powder form. Further, the powder was crushed with a ball mill to prepare fine powder of homogeneous size. Finally, the powder was sieved through 120 μM size sieve to get fine powder and stored in clean and dry glass bottles. The adsorbents were named MDP (majdool date pits) and SDP (sagai date pits), respectively.

### 2.5. H_2_O_2_ Treatment of Date Powders

Both the prepared MDP and SDP powder samples were treated with H_2_O_2_ (200 mL, 30% *v*/*v*) for the possible introduction of negative functionalities (-OH) onto the adsorbent surface. It also helps to decompose organic content, reduce the biomass recalcitrance and prevent cellulose degradation of the MDP and SDP powder [32]. This process was carried out by taking 10 g powders of each date type in a 100 mL beaker individually and 100 mL H_2_O_2_ solution and adding them into the beakers. Then, for proper mixing, a magnetic bar was placed inside each beaker, and the beakers were kept overnight in a magnetic stirrer at 125 rpm. Finally, the suspension was filtered through Buckner funnel using a vacuum pump and washed several times with DI water to remove any traces of H_2_O_2_. Then, the filtered powders were dried overnight with an oven at 90 °C. Then, they were taken out, allowed to cool and stored in clean glass bottles until further testing for metal ions adsorption. The treated materials were named TMDP and TSDP for majdool and sagai date pits, respectively.

### 2.6. Adsorption and Desorption Studies

Removal of heavy metal ions was carried out through batch adsorption studies. To do so, 0.05 g powder of each date’s seeds adsorbents were put into a previously cleaned and dry 100 mL Erlenmeyer flask, and 20 ppm (C_o_) metal solutions were added individually into the flask. The solutions in the flasks were kept in a shaker overnight with a speed of 100 rpm to allow equilibration. The mixtures were then filtered and stored in a cool and dry place. At equilibrium, the concentration of metal ions, C_e,_ was quantitatively determined by atomic adsorption spectroscopies, AAS (Thermo Scientific Evolution 600, Waltham, MA, USA).

The percentage adsorption (%) and metal ions removal capacity of the adsorbents at equilibration time (q_e_) and at arbitrary time t (q_t_) were calculated according to Equations (1)–(3) [32]:(1)Adsorption%=(Co−Ce)Co×100
(2)adsorption capacity at equilibrium (qe)=(Co−Ce)×Vm
(3)adsorption capacity at time t (qt)=(Co−Ct)×Vm
where C_o_, C_e_, and C_t_ are the initial, equilibrium and at-time-t concentrations, respectively (mg/L), of heavy metal solution; V is the volume (L) of heavy metal solution; and m is the mass (g) of adsorbents. The adsorption capacities, both (q_e_) and (q_t_), were calculated as mg/g.

The initial pH (pH_i_) of metal solutions of C_o_, 25 mg/L, was changed from 2 to 10 for the pH investigations. The effects of metal ion starting concentration (C_o_) on adsorption were investigated in the range of 20–100 mg/L at temperatures ranging from 293 to 323 K. The contact time (t) experiment for the target analytes adsorption at C_o_ of 25 mg/L were carried out at intervals ranging from 2 min to 24 h.

For desorption experiments, adsorbents (0.05 g) were initially saturated for 24 h with metal ions solutions (50 mL; 25 mg/L) individually. To remove unadsorbed traces of treated metal ions, saturated adsorbents were gently rinsed with DI water. Thereafter, the metal ions from saturated adsorbent samples were eluted using different eluents, including NaOH, HNO_3_, HCl, H_2_SO_4_ and CH_3_COOH solutions (50 mL; 0.1 mol/L). The concentration of metal ions in the eluent was quantitatively determined, and the corresponding percentage of desorption (% Desorption) was calculated as:

%Desorption = (Concentration of metal ion desorbed/Concentration of metal ions adsorbed on adsorbent) × 100

### 2.7. Thermodynamic Studies of Metal Ions Adsorptions

Thermodynamic parameters are very important for adsorption studies, indicating the spontaneity of the adsorption process. The negative value of Gibb’s free energy change (ΔG°) for a given temperature indicates the spontaneity of the adsorption process. Among the parameters, Gibb’s free energy change (ΔG°) were calculated using Equation (1)
ΔG° = −RT lnKa, (4)
while the change in enthalpy (ΔH°) and change in entropy (ΔS°) were determined using the equation,
ΔG° = ΔH° − TΔS°(5)
where R is the gas constant (8.314 J/mol K), T is the temperature in Kelvin (K) and Ka is the Langmuir constant [33].

## 3. Results and Discussion

### 3.1. Characterization

#### 3.1.1. Analysis of Adsorbent’s Surface and Pore Size

Surface morphology and physical properties of different as-prepared and treated adsorbents were investigated by SEM. Figure 2 shows the SEM images of MDP, TMDP, SDP and TSDP samples. The SEM image indicates the rough, highly porous and defined structure surface morphologies of the adsorbents suitable for metal adsorption. In addition, the SEM micrographs revealed that the treated date pits (TMDP, TSDP) were of higher porosity compared to the raw material (MDP, SDP). The Energy Dispersive X-ray Spectroscopy Analysis (EDX) was performed to determine the composition of the adsorbents, which confirms that the adsorbents consisted of carbon and negative heteroatoms such as oxygen and nitrogen. Upon H_2_O_2_ treatment, the adsorbents show a higher amount of oxygen, as expected (33 to 41%), indicating the conversion of DPs into negatively charged surfaces (Table 1).

The BET surface areas were determined for untreated and H_2_O_2_-treated date pit powder. The surface area for treated materials was found to increase from 2.94 to 9.97 m^2^/g in TMDP, while for TSDP, the areas changed from 2.80 to 9.46 m^2^/g. These changes in areas indicate the development of porosity during the H_2_O_2_ treatment of the as-prepared materials and enhancing their adsorption properties [32]. The adsorption average pore widths of both H_2_O_2_-treated and untreated dates pit powder were found to be in the range of 5–10 nm.

#### 3.1.2. Fourier-Transform Infrared Spectroscopic Analysis (FTIR)

Before and after metal ion adsorption, FTIR was used to better understand the adsorption mechanism for both as-prepared and treated adsorbents. FTIR is a useful tool for identifying the functional groups on the surface of the adsorbent and how they are affected by the experimental circumstances. RDP is a type of lignocellulosic material made up primarily of cellulose, hemicellulose, lignin and protein. Oxygen-rich components such as hydroxyl, ether and carbonyl functional groups are abundant in cellulose and hemicellulose. The ability of RDP to adsorb heavy metals is explained by the presence of these groups on its surface [34]. Figure 3 (upper panel) shows a broad peak in the 3230–3560 cm^−1^ range, indicating the presence of -OH, -NH or both, which is characteristic of lignocellulosic organic polymers [35]. The two peaks at 2921 and 2843 cm^−1^ indicate aliphatic C-H stretching vibrations. Peaks at 1739, 1602 and 1039 cm^−1^ show the presence of C=O (unconjugated carbonyl), C=C and C-O, respectively, with the small peaks between C=C and C-O referring to methyl group bending peaks [34,36]. The phenolic and C–O stretching vibrations of carboxylates produced a band at 1403–1432 cm^−1^. The bands between 1200 and 1350 cm^−1^ and 1000 and 1150 cm^−1^ were due to carboxylic acid and C–O–C vibrations, respectively. The presence of -OH polysaccharide groups is confirmed by a band at 500–700 cm^−1^. A minor shift (to higher wavenumbers) in peak position and increases in peak intensities were observed after Cr(III) and Cd(II) adsorption on adsorbents at peaks 3230–3560 cm^−1^,1739 cm^−1^ and 1039 cm^−1^. A minor shift (to higher wavenumbers) in peak position and increases in peak intensities were observed after Cr(III) and Cd(II) adsorption on adsorbents at peaks 3230–3560 cm^−1^, 1739 cm^−1^ and 1039 cm^−1^, which shows the involvement of the corresponding moieties in adsorption of metal ions. From these findings, it can be postulated that -OH or NH C=O and C-O functions were principally responsible for binding Cr(III) and Cd(II) over the surface via electrostatic interactions and lone-pair donation [32]. As a result, electrostatic interactions, as well as coordination-binding, could be the possible mechanism for metal adsorption.

### 3.2. Adsorption Properties

#### 3.2.1. Effect of pH

Adsorptions of Cr(III) and Cd(II) on MDP, TMDP, SDP and TSDP were studied for pH_i_ range: of 2.9 to 11.1, while pH higher than that was excluded to avoid any precipitation of metal ions as hydroxide species [37]. All the metal ions are cationic in solutions, and there is a possibility of competition in adsorption between the metal ions and the proton (H^+^) in the initial pH (pH_i_) range considered in this experiment. The adsorption capacities of Cr(III) ions were initially found to be increased and reached the maximum values of 158 mg/g and 192 mg/g at pH 7.8, while for Cd(II), it reached its highest values of 108 mg/g and 121 mg/g at pH 7.8, which was also true for MDP and TMDP, respectively. Similarly, adsorption capacities of Cr(III) were found to be increased from 56 to 132 mg/g and 86 to 193 mg/g for the pH increments of 2.9 to 6.5, while for Cd(II), the adsorption capacities were found to be maximum 94 mg/g and 148 mg/g at pH 7.8 for SDP and TSDP, respectively. The decrease in adsorption capacities was recorded for a further increase in pH (Figure 4). The adsorption maxima for both Cr(III) and Cd(II) were observed at pH values of 7.8, which were above the point of zero charges (pH_PZC_) for MDP, TMDP, SDP and TSDP (Figure 4). At pH 6.3 and 6.5, the pH_PZC_ value becomes zero for Cr(III) and Cd(II), respectively, indicating the neutral surface of the materials. Below and above this pHpzc value, the surfaces become positive and negatively charged, respectively. Both Cr(III) and Cd(II) are cationic; thus, at acidic conditions, it is possible that the surfaces of both TMDP and TSDP become excessively protonated, which inhibits the binding of metal ions on the adsorbent surfaces. The protonation decreases with the increase in pH (7.8), resulting in an increase in Cr(III) and Cd(II) binding over TMDP and TSDP. A drift in the pH graph was obtained above pH 7.8 during the study (Figure 4), suggesting the occurrences of neutralization of the adsorbents surface together with adsorption equilibrium.

#### 3.2.2. Effect of Initial Concentration and Temperature

Adsorption of Cr(III) (at C_o_: 10, 20, 30, 40 and 50 mg/L) on as-prepared and treated dates pit adsorbents (TMDP and TSDP) were studied at temperatures range of 20 to 50 °C to determine their corresponding equilibrium isotherms and presented in Figure 5. Initially, the vertical slopes of the plots represent a rapid increase in solid-phase concentrations of both Cr(III) and Cd(II) on the treated date pits with the increment of liquid-phase concentrations. The same trends with lower adsorption capacities were observed for the as-prepared adsorbents (MDP and SDP, not shown in the Figure). The correlation coefficient (R^2^) values for the graph C_e_ vs. q_e_ at different temperatures were found to be in the range of 0.8686 to 0.9955 for Cr(III), while they were in the range of 0.9814 to 0.9975 for Cd(II) adsorption. The increase in their respective concentration gradients might be acting as a driving force to overcome the resistance barrier mass transfer between solution/solid phases. Finally, the slopes of the plots started to decrease and become parallel to C_e_ (*x*-axis), indicating the saturation of both untreated and treated date pit adsorbent surfaces with Cr(III) adsorption. The adsorption of Cr(III) and Cd(II) on both the untreated and treated date pit adsorbents were increased with temperature, confirming the endothermic adsorption process, consistent with previous studies on metal ion adsorption [38]. At 293 K temperature, the adsorption capacities of Cr(III) and Cd(II) on TMDP were 55.3–217 and 15.3–120.0 mg/g, while on TSDP, the solid-phase concentration ranges (q_e_) were 53.2–199.7 and 11.5–115.0 mg/g respectively. The regular increments in the q_e_ values were noticed with increasing temperature. At temperatures 303, 313 and 318 K, the solid phase concentration ranges of Cr(III) on TMDP were 58.3–245.4, 59.9–269.9 and 60.1–278.9, while on TSDP the ranges of q_e_ were 58.2–201.5, 59.9–209.9 and 60.1–213.9 mg/g, respectively. Similarly, for Cd(II), the solid phase concentration ranges on TMDP were 19.8–132.0, 20.9–140.0 and 21.5–141.8, while on TSDP, the q_e_ was found in the ranges of 16.2–122.0, 18.2–135.0 and 21.2–139.18 mg/g at temperatures of 303, 313, and 318 K, respectively.

#### 3.2.3. Effects of Contact Time

Adsorption of Cr(III) and Cd(II) on TMDP and TSDP as a function of contact time for different concentrations of the metal ions was tested from 5 to 300 min, and Figure 6 illustrates the effects of contact time on adsorption of both the metal ions. Slow adsorption of both the metal ions was observed on TMDP and TSDP and reached an equilibrium in 300 min (5 h). The R^2^ values for the graph of time (min) vs. Qe (mg/g) at various concentration levels were found to be in the range of 0.7644 to 0.9881 for Cr(III) and 0.8995 to 0.9477 for Cd(II) adsorption. The equilibrium adsorption capacity (Qe) for Cr(III) on TMDP was 190, 281 and 451 mg/g, while for Cd(II) ions, the Qe was 121, 211 and 371 mg/g for the concentrations of 20, 30 and 50 ppm, respectively. Similarly, the adsorption capacity (Qe) for Cr(III) on TSDP was 140, 229 and 359 mg/g, while for Cd(II) ions, the Qe was 109, 198 and 298 mg/g for the concentrations of 20, 30 and 50 ppm, respectively. Lower adsorption capacities were noticed for as-prepared materials, MDP and SDP (not shown in the Figure).

#### 3.2.4. Adsorption Modeling

##### Equilibrium Isotherm

Two-parameter analysis during the study and Langmuir and Freundlich [39] isotherm models in linear form were used.

Langmuir isotherm model in non-linearized and linearized form is expressed as
(6)qe=qmkLCe1+kLCe
(7)Ceqe=1kLqm+1qm×Ce
where q_m_ (mg/g) and K_L_ (L/mg) are the constants for maximum monolayer adsorption capacity and a constant related to the heat of adsorption, respectively.

Freundlich isotherm in linear and non-linear form is expressed as [39]
(8)qe= KF×Ce1/n
(9)logqe=logKF+1n logCe
where K_F_ ((mg/g) (L/mg)^(1/n)^) and n are the Freundlich constants related to bonding energy and deviation in adsorption from linearity, respectively. If n *=* 1 (linear adsorption process), n < 1 (chemical adsorption process), n > 1 (physical adsorption process).

The relevant parameters for Cr(III) and Cd(II) adsorption on TMDP and TSDP at various temperatures are shown in Table 2. Regression coefficient (R^2^) values for Cr(III) and Cd(II) adsorption on both TMDP and TSDP at varied temperatures were comparatively higher for Langmuir isotherm model, agreeing well with previously reported results of metal ion adsorption [32,40]. In addition, the applicability of the Langmuir model hints towards a monolayer metal ions coverage over adsorbents at varied temperatures. Maximum monolayer adsorption capacity (q_m_) for Cr(III) increased from 400 (400/51.99 = 7.693 mmol/g) to 1428 mg/g (1428/51.99 = 27.46 mmol/g) for TMDP and 388 mg/g (388/51.99 = 7.463 mmol/g) to 1182 mg/g (1182/51.99 = 27.735 mmol/g) for TSDP materials, while the temperature was increased from 293 K to 323 K. The maximum monolayer adsorption for Cd(II) for Cd(II) increased from 352 mg/g (352/112.4 = 3.132 mmol/g) to 1302 mg/g (1302/112.4 = 11.584 mmol/g) for TMDP and 332 (332/112.4 = 2.954 mmol/g) to 1182 mg/g (1182/112.4 = 10.516 mmol/g) for TSDP materials, with increasing temperature from 293 to 323 K. The increase in temperature may have increased the frequency of collisions between the adsorbents and Cr(III) and Cd(II), resulting in enhanced adsorption. Additionally, the rise in temperature might have ruptured the surface bonds of the materials, exposing the active site and thus enhancing metal ion adsorption. The magnitudes of the separation factor (K_L_) for both Cr(III) and Cd(II) adsorption on TMDP and TSDP at the studied temperature ranges were in the range between 0 and 1, indicating a favorable adsorption process. The magnitude of K_F_ increased with temperature, as for the intensification of metal ions and adsorbents interaction with temperature, and the n > 1 values confirm the physical adsorption process (Table 2).

### 3.3. Adsorption Kinetics

Pseudo-first-order [41], and pseudo-second-order [42] models in linear form were used on adsorption kinetic data, and the models in linearized forms are as follows:(10)log(qe1 − qt)=logqe−k12.303×t
(11)tqt=1k2qe22+1qt×t
where, the adsorption capacities for the pseudo-first-order model and pseudo-second-order model at equilibria and at time t are denoted by q_e1_, q_e2_ and q_t_, respectively, while k_1_ and k_2_ are the pseudo-first-order and pseudo-second-order rate constants, respectively.

For Cr(III) and Cd(II) adsorption on MDP, TMDP, SDP and TSDP, the obtained kinetic parameters are shown in Table 3. Higher R^2^ (closer to unity) supports pseudo-second-order kinetic model fitting to both the metal ions (II) adsorption data in all the tested adsorption systems. Similarly, the closeness of the experimental adsorption capacities (q_e,exp_*)* to computed adsorption capacities (q_e2,cal._) validate the applicability of the aforementioned pseudo-second-order model to the data. Ghobadi et al. [43] reported similar results for La(III) and Ce(III)) adsorption on MnFe_2_O_4_-graphene nanoparticles.

### 3.4. Adsorption Thermodynamics

Thermodynamic parameters relating to Cr(III) and Cd(II) adsorption on various adsorbents, such as standard changes of entropy (ΔS°), enthalpy (ΔH°) and free energy (ΔG°), are calculated using van Hoff plots [44] and are shown in Table 4. The adsorption of Cr(III) and Cd(II) on MDP, TMDP, SDP and TSDP was endothermic, as evidenced by positive ΔH° values. The ΔS° values for Cr(III) and Cd(II) adsorption on MDP, TMDP, SDP and TSDP were all positive, indicating that the adsorption process was random as a result of energy redistribution between the metal ions and the adsorbent [45]. Furthermore, the increase in the degree of freedom of the adsorbed metal ions is confirmed by positive ΔG° values. For all types of adsorbents, the ΔG° values for both Cr(III) and Cd(II) adsorption were negative, indicating that the adsorption process was spontaneous. With increasing temperature, much higher negative ΔG° values suggest that the adsorption process tended toward spontaneity [32].

### 3.5. Elution and Regeneration Studies

The number of eluents was used to test the Cr(III)- and Cd(II)-saturated adsorbents for elution investigations. The elution of both metal ions from all saturated adsorbents was found to be minimal (2.7–3.8%) with 0.1 mol/L NaOH; however, with 0.1 M HCl, a sufficiently higher amount of metal ions was desorbed for all adsorbents (83.0–97.2%) (Figure 7). The elution of both Cr(III) and Cd(II) with eluents followed the order: 0.1 mol/L NaOH < 0.1 mol/L CH_3_COOH < 0.1 mol/L H_2_SO_4_ < 0.1 mol/L HNO_3_ < 0.1 mol/L HCl. Thus, the maximum Cr(III) ions elution achieved with HCl (0.1 mol/L) were (97.2%), while it was 94.0% for Cd(II) ions. The higher desorption of both metal ions with comparatively strong acid indicated an ion-exchange process, which might govern metal ions binding on adsorbents surface.

Following the optimization of HCl (0.1 mol/L) for Cr(III) and Cd(II) from saturated MDP, TMDP, SDP and TSDP, regeneration tests were conducted to determine the reusability efficiency of the adsorbents. A drastic fall in Cr(III) (85.3–88.2%) and Cd(II) (81.8–86.8%) adsorption on MDP, TMDP, SDP and TSDP were observed after the second regeneration steps. The distortion of surface morphology during repeated adsorption and elutions of Cr(III) and Cd(II) was a probable mechanism behind a drop in metal ion adsorption. As a result, MDP, TMDP, SDP and TSDP could be effectively reutilized for Cr(III) and Cd(II) adsorption with no significant adsorption potential loss during initial regeneration.

There is inadequate research using date seed powders as adsorbents for the removal of heavy metals. Few research studies have used date stones, raw date pits and burnt date pits as potential adsorbents rather dates pit powders for the removal of heavy metals [46,47], while, an enhanced adsorption of heavy metals was reported with the burning of the date seeds. Mohamed et al. [48] reported Cu(II) adsorption using ajwa date pits powder, and the carbonized date seeds for the removal of Cu(II) from wastewater was demonstrated by others [49]. Our results for Cr(III) and Cd(II) removal using MDP, SDP, TMDP and TSDP were not significantly different from the reported findings in terms of heavy metal adsorption; only the preparation of the adsorbents was slightly different. However, in the current research, both the untreated and H2O2 treated materials showed excellent adsorption efficiency for Cr(III) and Cd(II) ions.

A series of batch experiments was performed to optimize the adsorbent dosage, solution pH, contact time and initial adsorbate concentrations, and the metal ions were determined using atomic adsorption spectrometry (AAS). The maximum adsorption capacity for all the metal ions was noted at pH 7.8. The obtained results were found to be best fitted with the Langmuir isotherm model, where the maximum adsorption capacities (monolayer) at 323 K were 1428.5 mg/g and 1302.0 mg/g (TMDP) and 1228.5 mg/g and 1182.0 mg/g (TSDP) for Cr(III) and Cd(II), respectively. The efficiencies of all the adsorbents—MDP, SDP, TMDP and TSDP—were compared, and it was found that for both Cr and Cd elements, the adsorption capacities of TMDP and TSDP were much higher compared to untreated materials (MDP and SDP). However, a slightly higher adsorption capacity of TMDP was noticed compared to TSDP, and the most efficiently removed metal ion was Cr(III), then Cd(II), for all the adsorbents. This could be attributed to the surface morphology and pore size of the adsorbents. The higher pore size and surface functionality of the H_2_O_2_ treated material compared to MDP and SDP were indicated by FTIR, BET, and SEM results.

## 4. Conclusions

In conclusion, optimized MDP, TMDP, SDP and TSDP exhibited excellent adsorption potential for Cr(III) and Cd(II) in their respective adsorption systems. The H_2_O_2_-treated materials showed better adsorption capacities compared to as-prepared materials. However, better adsorptions of Cr(III) metal were noticed compared to Cd(II) in all the tested adsorbents. The obtained R^2^ for C_e_ vs. Q_e_ graph at various temperatures was in the range of 0.8686 to 0.9975 for both Cr(III)and Cd(II) adsorption. Similarly, the R^2^ for the Time (min) vs. Qe (mg/g) graph was obtained in the range of 0.7644 to 0.9881 for both metal ions at varying concentrations. The adsorptions of both metal ions on MDP, TMDP, SDP and TSDP were endothermic, and a pseudo-second-order kinetics model is better suited in all cases. Langmuir isotherm model showed slightly better results than the Freundlich model for the untreated and treated date pits, indicating monolayer coverage. Elution experiments showed maximum Cr(III) and Cd(II) recovery with 0.1 mol/L HCl. The obtained adsorption and regeneration results indicate that the synthesized adsorbent materials could be useful for the adsorption of heavy metals from the aqueous system.

## Figures and Tables

**Figure 1 molecules-26-03718-f001:**
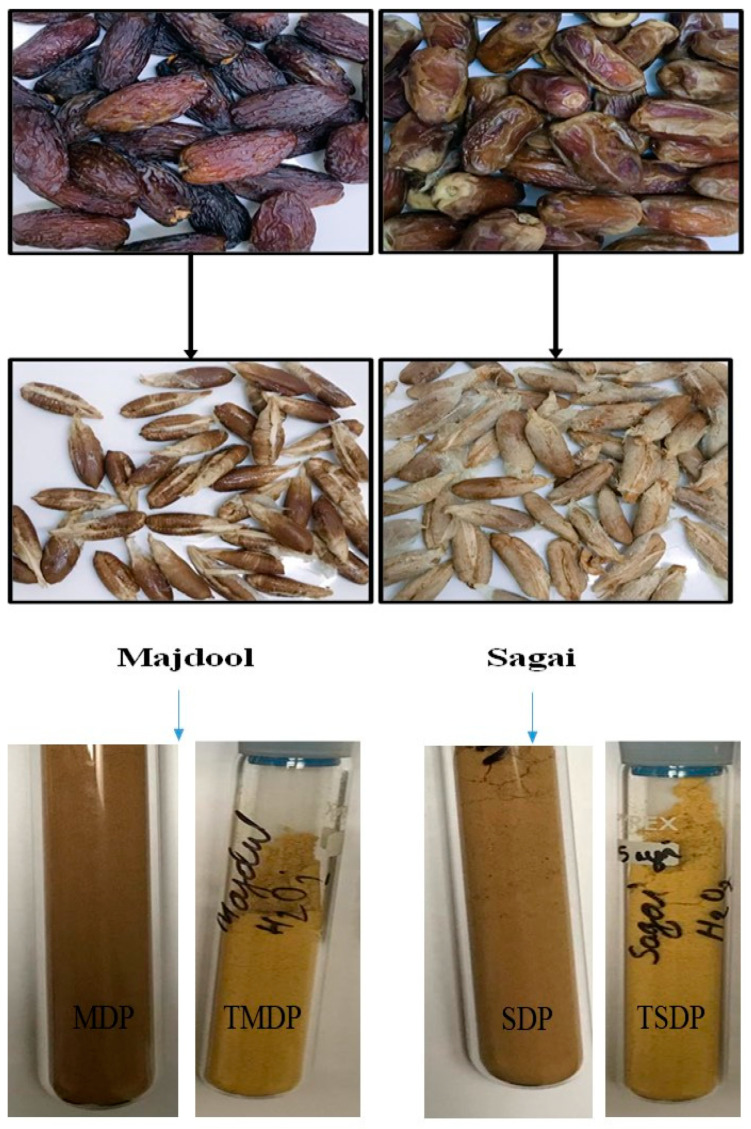
The picture of date samples, seeds and prepared adsorbents (MDP, TMDP, SDP and TSDP).

**Figure 2 molecules-26-03718-f002:**
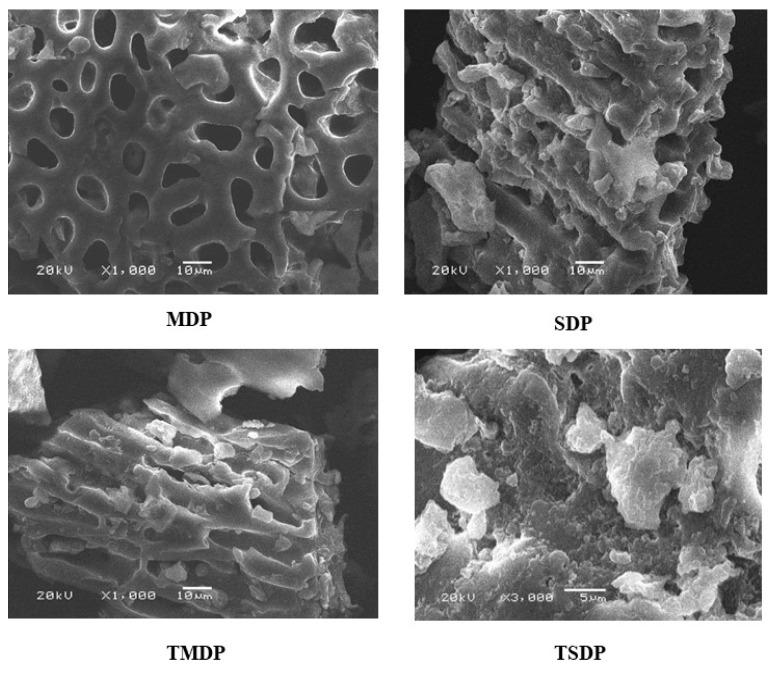
SEM of MDP, SDP and TMDP of 1000× and 10 μm and TSDP of 3000× magnification and 5 μm diameter.

**Figure 3 molecules-26-03718-f003:**
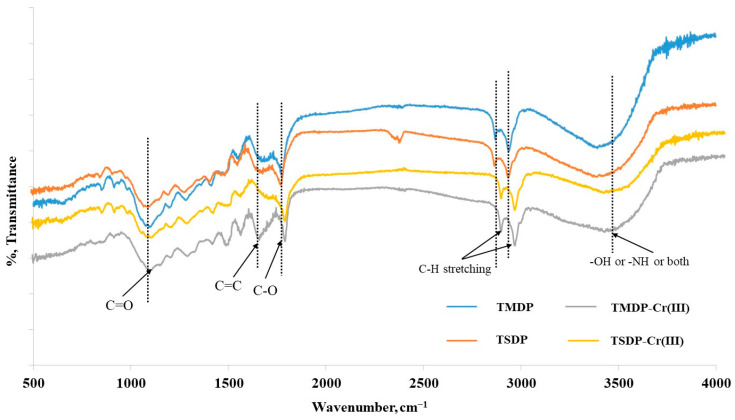
FT-IR of TMDP, TMDP-Cr(III), TSDP and TSDP-Cr(III). (Experimental conditions: C_o_: 25 mg/L; m: 0.005 g; V: 0.05 L; T: 25 °C; contact time: 5 h; agitation speed: 100 rpm).

**Figure 4 molecules-26-03718-f004:**
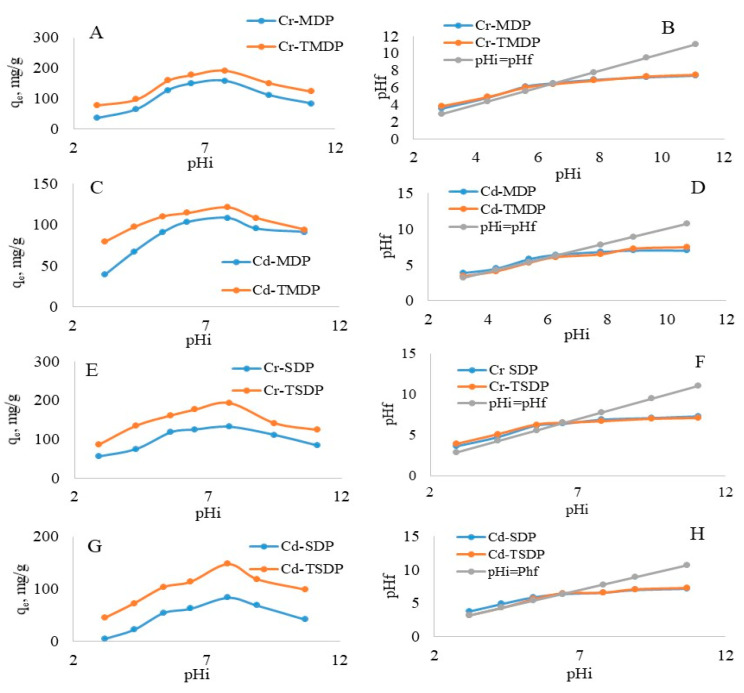
Equilibrium adsorbed concentration (qe) as a function of initial pH (pHi) (**A**,**C**,**E**,**G**), and pHi versus final pH (pHf) plot (**B**,**D**,**F**,**H**) for Cr(III) and Cd(II) adsorption onto MDP, TMDP, SDP and TSDP. (C_o_: 25 mg/L; m: 0.005 g; V: 0.05 L; T: 25 °C; contact time: 5 h; agitation speed: 100 rpm).

**Figure 5 molecules-26-03718-f005:**
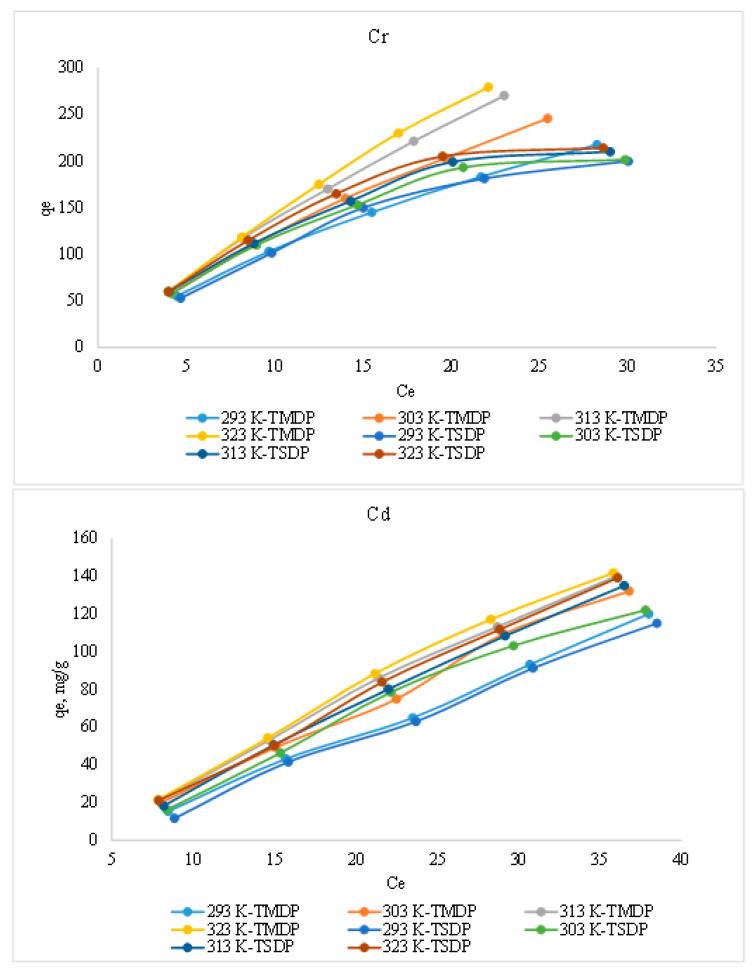
Effect of initial concentration of the adsorbates and the temperature (experimental conditions: C_o_: 25 mg/L; m: 0.005 g; V: 0.05 L; contact time: 5 h; agitation speed: 100 rpm).

**Figure 6 molecules-26-03718-f006:**
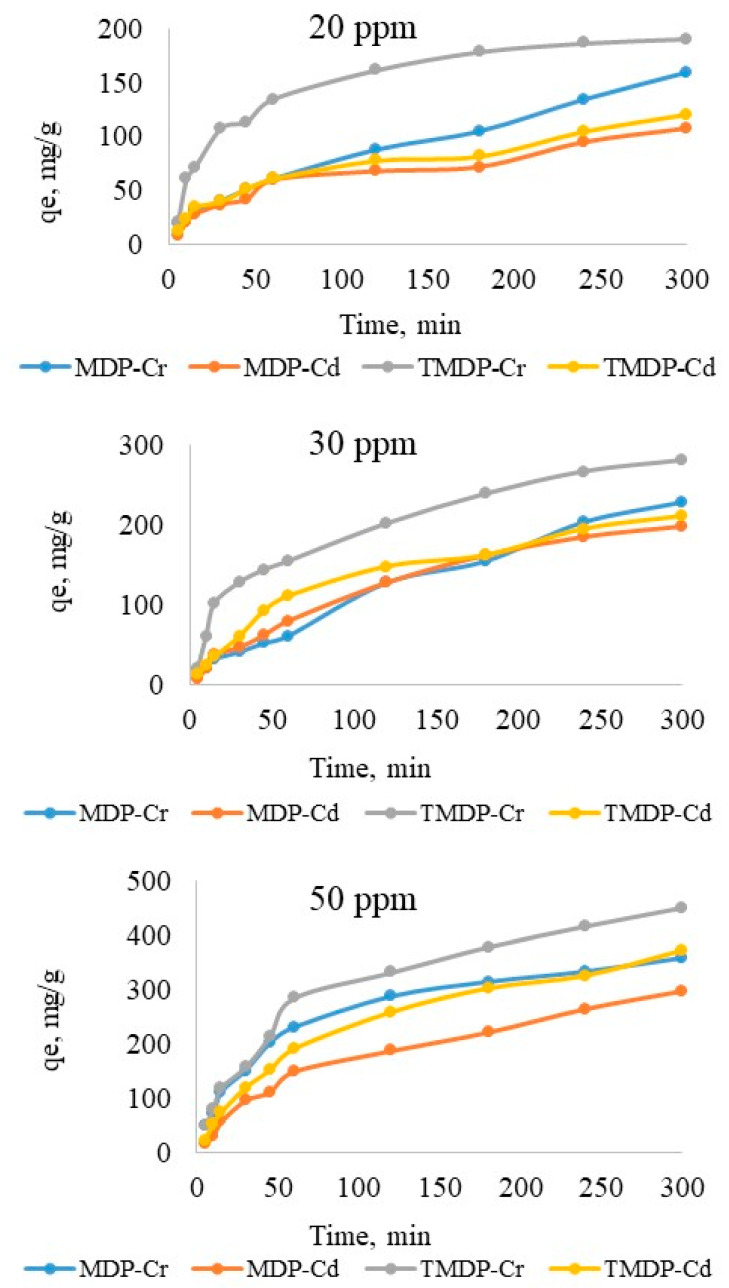
Effect of contact time, at various concentrations of the adsorbates (Experimental conditions: C_o_: 25 mg/L; m: 0.005 g; V: 0.05 L; T: 25 °C; agitation speed: 100 rpm).

**Figure 7 molecules-26-03718-f007:**
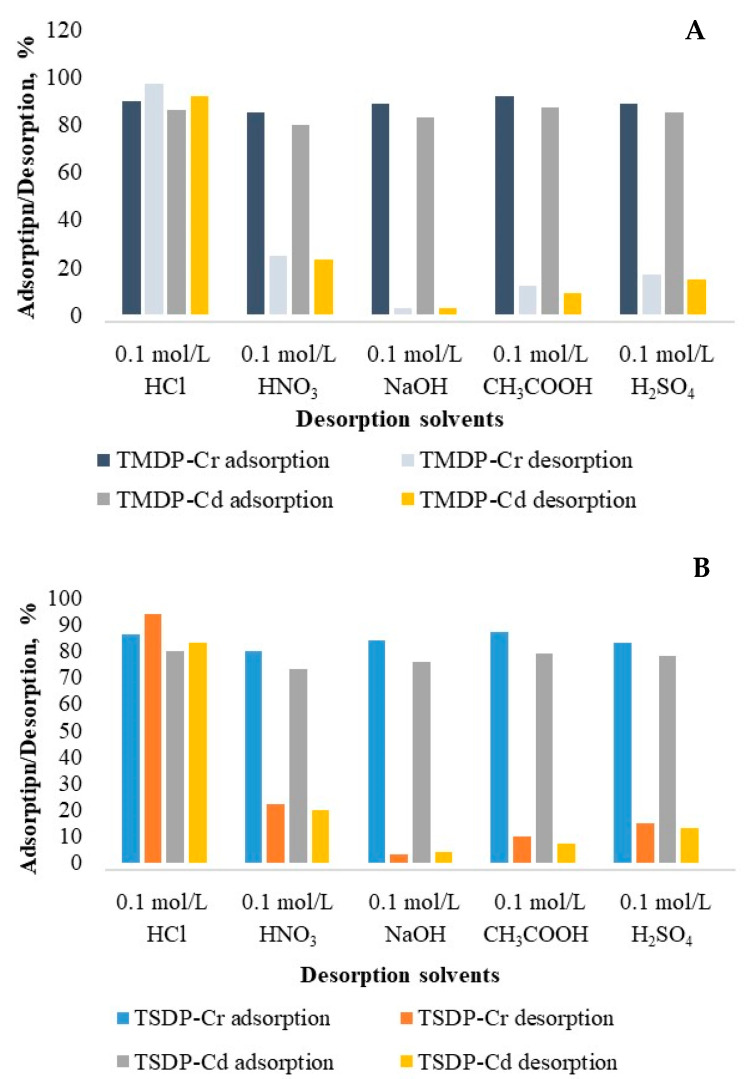
Elution of Cr(III) and Cd(II) from saturated adsorbents, TMDP (**A**) and TSDP (**B**). (C_o_: 25 mg/L; m: 0.05 g; V: 0.05 L; T: 25 °C; contact time: 5 h; agitation speed: 100 rpm).

**Table 1 molecules-26-03718-t001:** EDX analysis of the as-prepared (MDP, SDP) and treated (TMDP, TSDP) materials.

Element	(keV)	Mass%							
		MDP	Error	SDP	Error	TMDP	Error	TSDP	Error
C··K	0.277	40.11	0.10	40.10	0.11	37.22	0.10	37.80	0.11
N··K	0.392	26.64	0.28	27.60	0.25	21.4	0.09	22.30	0.06
O··K	0.525	33.26	0.08	32.30	0.07	41.38	0.06	39.90	0.09
Total		100		100		100		100	

**Table 2 molecules-26-03718-t002:** Isotherm parameters for the adsorption of Cr(III) and Cd(II) on TMDP and TSDP.

Isotherm	Temperature, K							
	293		303		313		323	
	Cr(III)	Cd(II)	Cr(III)	Cd(II)	Cr(III)	Cd(II)	Cr(III)	Cd(II)
TMDP								
Langmuir								
q_m_ (mg/g)	400	352	666.67	602	1000	908	1428.5	1302
K_L_ (L/mg)	0.03592	0.02920	0.03727	0.03870	0.05744	0.04344	0.07599	0.05494
R^2^	0.9929	0.9881	0.9952	0.9897	0.9963	0.9901	0.9981	0.9932
Freundlich								
KF (mg/g)	18.5642	12.5642	28.5642	19.258	46.5142	36.2548	68.56142	56.258
(L/mg)^1/n^								
n	1.3821	1.0821	1.6821	1.3021	1.8211	1.5821	1.9821	1.7821
R^2^	0.9763	0.9485	0.9805	0.9622	0.9884	0.9715	0.9887	0.9810
TSDP								
Langmuir								
q_m_ (mg/g)	388	332	606.67	552	910	808	1228.5	1182
K_L_ (L/mg)	0.03222	0.02520	0.03327	0.03071	0.05044	0.04004	0.07219	0.0509
R^2^	0.9909	0.9289	0.9921	0.9611	0.9938	0.9822	0.9941	0.9938
Freundlich								
KF (mg/g)	14.2125	8.2314	22.0213	15.1254	37.8541	28. 8254	56.2242	42.1258
(L/mg)^1/n^								
n	1.0132	1.0021	1.2168	1.1021	1.6211	1.3021	1.7621	1.2154
R^2^	0.9521	0.8257	0.9644	0.8666	0.9788	0.8801	0.9801	0.9002

**Table 3 molecules-26-03718-t003:** Kinetic parameters for the adsorption of Cr(III) and Cd(II) on various DP adsorbents (initial concentration, C_o_ for Cr(III) and Cd(II) was 20 mg/L).

Kinetics Model	MDP		TMDP		SDP		TSDP	
	Cr(III)	Cd(II)	Cr(III)	Cd(II)	Cr(III)	Cd(II)	Cr(III)	Cd(II)
q_e,exp_ (mg/g)	159	108	191	121	140	109	190	120
Pseudo-first-order								
q_e1,cal_ (mg/g)	132	85	151	101	109	79	148	101
K_1_(1/min)	0.0064	0.0059	0.0062	0.0061	0.0060	0.0058	0.0068	0.0063
R^2^	0.9787	0.9688	0.9822	0.9588	0.9666	0.9589	0.9787	0.9689
Pseudo-second-order								
q_e2,cal_ (mg/g)	162	110	194	123	144	111	195	122
k_2_ (g/mg.min)	0.0004	0.0009	0.0002	0.0006	0.0006	0.0005	0.0008	0.0007
R^2^	0.9907	0.9818	0.9877	0.9900	0.9811	0.9901	0.9878	0.9719

**Table 4 molecules-26-03718-t004:** Thermodynamic parameters for the adsorption of Cr(III) and Cd(II) on various DP adsorbents (initial concentration, C_0_ for Cr(III) and Cd(II) was 20 mg/L).

Adsorbents ↓	ΔH° (kJ/mol)	ΔS° (J/mol-K)	ΔG° (kJ/mol)			
Temperature →			293 K	303 K	313 K	323 K
MDP	6154	20.77	−113.79	−430.13	−644.08	−802.54
TMDP	8511	29.77	−153.79	−539.13	−877.08	−912.54
SDP	4254	15.22	−103.79	−295.13	−523.08	−912.54
TSDP	7588	27.66	−143.79	−509.13	−697.08	−832.54

## Data Availability

The study did not report any data.

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
