# Peer review of "Removal of Chromium(III) and Cadmium(II) Heavy Metal Ions from Aqueous Solutions Using Treated Date Seeds: An Eco-Friendly Method"

_molecules, 2021, doi:10.3390/molecules26123718_

Round 1
Reviewer 1 Report
Authors present a work in which they prepare low-cost adsorbents including raw date pits and chemically treated date pits to investigate the adsorption of Cr(III) and Cd(II) ions from wastewater.
It results interesting that they employ date sheets, very abundant in the country of the authors, for clean and green purposes, increasing the added value of organic residues.
Due to the presence of various functional groups such as carboxylic acid, ester, carboxylate, hydroxyl, phenolic and amino, materials prepared by authors can serve as adsorption sites for the above cited heavy metal ions.
They start with a presentation in which they properly describe the state of art, with many references for the reader, most of them actual and new.
Employed materials and method are described then. At materials, the use of spatula and filtering paper is pointed out as material...I find that is not necessary, it is enough by describing the reactants. The preparation of biosorbent is properly explained and shown. The image shown in Figure 1 should be improved, the quality is low. Adsorption and desorption studies carried out are well presented and explained, with the use of expresions for the calculations. They employed the abbreviations hrs for the time in hours, it should be changed by "h".
Regarding results presented, SEM images are of good quality and show properly that described in the text by the authors. For functional group characterization, they present FTIR spectra of samples in Figure 3. Even in the interpretation is correct, in my opinion the spectra can be improved and some bands indicated by arrows, for a better understanding.
Graphs related to absorption study should also be improved, the quality is quite low and it difficults visualization in a correct way. The interpretation, however, seems to be correct. The effects of initial concentration and temperature, together with that of contact time are well presented and analyzed, but again the figures can be improved in quality. Modelization and kinetics are presented in a proper way and well performed, but again, the equations presented should be rewritten for adequate to the rest of the text, as they are different in letter and format. Errors for Table 1 and Table 2 would also be interesting. Some data should be rewritten to the format "0." instead of ".0"
Conclusions extracted are in agreement with presented data, so after some corrections the manuscript could be published.
Author Response
Comments and Suggestions for Authors
Authors present a work in which they prepare low-cost adsorbents including raw date pits and chemically treated date pits to investigate the adsorption of Cr(III) and Cd(II) ions from wastewater.
It results interesting that they employ date sheets, very abundant in the country of the authors, for clean and green purposes, increasing the added value of organic residues.
Due to the presence of various functional groups such as carboxylic acid, ester, carboxylate, hydroxyl, phenolic and amino, materials prepared by authors can serve as adsorption sites for the above cited heavy metal ions.
They start with a presentation in which they properly describe the state of art, with many references for the reader, most of them actual and new.
Ans: Thank you so much for appreciating our work.
Employed materials and method are described then. At materials, the use of spatula and filtering paper is pointed out as material...I find that is not necessary, it is enough by describing the reactants. The preparation of biosorbent is properly explained and shown. The image shown in Figure 1 should be improved, the quality is low. Adsorption and desorption studies carried out are well presented and explained, with the use of expresions for the calculations. They employed the abbreviations hrs for the time in hours, it should be changed by "h".
Ans: All of the corrections have been made accordingly in the text.
Regarding results presented, SEM images are of good quality and show properly that described in the text by the authors. For functional group characterization, they present FTIR spectra of samples in Figure 3. Even in the interpretation is correct, in my opinion the spectra can be improved and some bands indicated by arrows, for a better understanding.
Ans: The spectra have been corrected accordingly in the revised manuscript.
Graphs related to absorption study should also be improved, the quality is quite low and it difficults visualization in a correct way. The interpretation, however, seems to be correct. The effects of initial concentration and temperature, together with that of contact time are well presented and analyzed, but again the figures can be improved in quality. Modelization and kinetics are presented in a proper way and well performed, but again, the equations presented should be rewritten for adequate to the rest of the text, as they are different in letter and format. Errors for Table 1 and Table 2 would also be interesting. Some data should be rewritten to the format "0." instead of ".0"
Ans: All of the corrections have been made accordingly in the text
The conclusions extracted are in agreement with the presented data, so after some corrections, the manuscript could be published.
Ans: All of the corrections have been made accordingly in the revised manuscript.
Reviewer 2 Report
- In line 11 it should be "the aim of the research was ..."
- The aim of the work is not very precise in lines 68-76. This paragraph should precisely describe the aim of the experiment being conducted and the novelty of this research.
- In formulas 1, 2 and 3 (lines between 147 and 148), we do not give units for “Adsorption”. This should be stated in the legend for the formulas.
- The relationships concerning Cr and Cd shown in Figure 5 should be interpreted statistically. I propose to provide the correlation coefficient rx,y. Statistical analysis is now necessary to understand the effect (dependence) of the dependent variable on the independent variable. At work, it will be a valuable supplement to the interpretation of the results.
- The same remark applies to the relationships shown in Figure 6. A statistical interpretation of the effect of Time (min) on Qe (mg/g) should be provided.
- Chapter 3 The authors called "RESULTS AND DISCUSSION". This chapter only describes the research results, but there is no discussion. The presented research results must be discussed with the reports described in the scientific literature of other scientists. This chapter must be completed with a discussion of the results.
- The chapter "Conclusions" should be supplemented with the statistical interpretation suggested in points 5 and 6.
In my opinion the experiment conducted by the authors of this publication is scientifically and cognitively interesting. However, the publication must be supplemented with comments included in points 1-7. The current version of the article is not suitable for publication.
Author Response
Comments and Suggestions for Authors
- In line 11 it should be "the aim of the research was ..."
Ans: Corrected in the text.
- The aim of the work is not very precise in lines 68-76. This paragraph should precisely describe the aim of the experiment being conducted and the novelty of this research.
Ans: In the present work, we have prepared low-cost adsorbents from date pits and applied them for the removal of potential heavy metals (cadmium and chromium) from aqueous solutions. The date seeds are abundantly available in Saudi Arabia, and the presence of various functional groups such as carboxylic acid, ester, carboxylate, hydroxyl, phenolic and amino in the materials can be serve them as potential adsorbents for the removal of heavy metal ions. The negative (–OH) surface functionalities of the as prepared date pit powders were enhance by treating them with hydrogen peroxide overnight on a magnetic stirrer. The as prepared and treated samples were characterized using SEM, BET, and FT-IR techniques, which confirms the excellent surface morphology and particle size of the adsorbent materials. The excellent adsorption capacities were obtained for the H2O2 treated date pits adsorbents and thus it can be assumed that the materials could be useful for the removals of toxic metals (Cr(III) and Cd(II)) from aqueous solutions.
- In formulas 1, 2 and 3 (lines between 147 and 148), we do not give units for “Adsorption”. This should be stated in the legend for the formulas.
Ans: Corrected in the text.
- The relationships concerning Cr and Cd shown in Figure 5 should be interpreted statistically. I propose to provide the correlation coefficient rx,y. Statistical analysis is now necessary to understand the effect (dependence) of the dependent variable on the independent variable. At work, it will be a valuable supplement to the interpretation of the results.
Ans: The correlation coefficient (R2) values for the graph Ce vs qe at different temperatures were found to be in the range of 0.8686 to 0.9955 for Cr(III), while it was in the range of 0.9814 to 0.9975 for Cd(II) adsorption.
- The same remark applies to the relationships shown in Figure 6. A statistical interpretation of the effect of Time (min) on Qe (mg/g) should be provided.
Ans: The R2 values for the graph Time (min) vs Qe (mg/g) at various concentration levels were found to be in the range of 0.7644 to 0.9881 for Cr(III) and 0.8995 to 0.9477 for Cd(II) adsorption.
- Chapter 3 The authors called "RESULTS AND DISCUSSION". This chapter only describes the research results, but there is no discussion. The presented research results must be discussed with the reports described in the scientific literature of other scientists. This chapter must be completed with a discussion of the results.
Ans: The below discussion has been added in the manuscript.
3.2.5 Discussion
There is inadequacy of research using date seed powders as adsorbents for heavy metals removal. Few researches were used date stones, raw date pits and burnt date pits as potential adsorbent rather dates pit powders for removal of heavy metals [46, 47] and they reported enhance adsorption of heavy metals on the burning of the date seeds. Mohamed et al [48] reported the Cu(II) adsorption using ajwa date pits powder and the carbonized date seeds for the removal of Cu(II) from wastewater was demonstrated by other workers [49]. Our results for Cr(II) and Cd(II) removal using MDP, SDP, TMDP and TSDP were not too different from the reported findings in terms of heavy metal adsorption, only the preparation of the adsorbents were of slight alteration. However, in the current research, both the untreated and H2O2 treated materials showed excellent adsorption efficiency towards Cr(III) and Cd(II) ions.
- The chapter "Conclusions" should be supplemented with the statistical interpretation suggested in points 5 and 6.
Ans: The conclusion has been revised accordingly in the text.
In my opinion the experiment conducted by the authors of this publication is scientifically and cognitively interesting. However, the publication must be supplemented with comments included in points 1-7. The current version of the article is not suitable for publication.
Ans: Thank you so much for appreciation. We have addressed all of the comments and suggestions raised by the learned reviewer.
Reviewer 3 Report
Heavy metal pollution is a serious environmental issue. This article focuses the Cr and Cd removal using treated date seeds, it is an eco-friendly method, so it is very interesting and important. Authors have done detailed research experiments, and get lots of valued data. But it need some revision before published in this journal.
Abstract: lack of data, please add some important data, for example maximum adsorption capacities etc;
Introduction: Authors have not explained why they modified the date seeds by H2O2, this is the basic innovation point of this article, so please add it.
Materials and Methods:
Line 92 MDP, SDP TMDP TSDP, please give the full name when use it the first time.
Line 121 please check the words “MDP, SDP TMDP TSDP” in the right bottles.
Line 128, “10 gm powders”; Line 138, “0.05gram”; please check the units, and keep them uniform.
Line 143, “by with” please check.
Line 149, “dye” what is dye? Please check.
Results and discussion:
Line 220-225, this paragraph should be edited again, please explain clearly what have changed, and why it changed.
Line 256 Figure 4 B, please check “CXR-TMDP”
All the tables should be the three line table, keep them uniformly
Line 376, “regenaration” should be “regeneration”
Line 386-393, please add figure to explain the regeneration studies, it will be clearly.
References: line 416 “chemosphere” and line 461 “chemosphere”, please keep the name of the journal the same typeface.
Author Response
Comments and Suggestions for Authors
Heavy metal pollution is a serious environmental issue. This article focuses the Cr and Cd removal using treated date seeds, it is an eco-friendly method, so it is very interesting and important. Authors have done detailed research experiments, and get lots of valued data. But it need some revision before published in this journal.
Abstract: lack of data, please add some important data, for example maximum adsorption capacities etc;
Ans: The data have been added to the abstract according to the honorable reviewer suggestion.
Introduction: Authors have not explained why they modified the date seeds by H2O2, this is the basic innovation point of this article, so please add it.
Ans: In the present work, we have prepared low-cost adsorbents from date pits and applied them for the removal of potential heavy metals (cadmium and chromium) from aqueous solutions. The date seeds are abundantly available in Saudi Arabia, and the presence of various functional groups such as carboxylic acid, ester, carboxylate, hydroxyl, phenolic and amino in the materials can be serve them as potential adsorbents for the removal of heavy metal ions. The negative (–OH) surface functionalities of the as prepared date pit powders were enhance by treating them with hydrogen peroxide overnight on a magnetic stirrer. The as prepared and treated samples were characterized using SEM, BET, and FT-IR techniques, which confirms the excellent surface morphology and particle size of the adsorbent materials. The excellent adsorption capacities were obtained for the H2O2 treated date pits adsorbents and thus it can be assumed that the materials could be useful for the removals of toxic metals (Cr(III) and Cd(II)) from aqueous solutions.
Materials and Methods:
Line 92 MDP, SDP TMDP TSDP, please give the full name when use it the first time.
Ans: We have corrected the same throughout the manuscript.
Line 121 please check the words “MDP, SDP TMDP TSDP” in the right bottles.
Ans: We have corrected the figure for better understanding.
Line 128, “10 gm powders”; Line 138, “0.05gram”; please check the units, and keep them uniform.
Ans: We have corrected the unit as ‘g’ only.
Line 143, “by with” please check.
Ans: Corrected to ‘by’.
Line 149, “dye” what is dye? Please check.
Ans: We are sorry for the typos, and corrected accordingly in the manuscript.
Results and discussion:
Line 220-225, this paragraph should be edited again, please explain clearly what have changed, and why it changed.
Ans: We have corrected the text as mentioned here.
After Cr(III) and Cd(II) adsorption on adsorbents, a slight shift (to the higher wavenumbers) in the peak positions and increases in peak intensities at peaks 3230 – 3560 cm-1 range, 1739 cm-1 and 1039 cm-1 were observed. These indicate the participation of the corresponding moieties in the adsorption of metal ions. From these results, it could be postulated that –OH or NH C=O and C-O functionalities were primarily responsible for binding Cr(III) and Cd(II) over the surface through electrostatic forces and lone pair donation [32]. Therefore, both electrostatic interactions and coordination binding might be the possible adsorption mechanisms.
Line 256 Figure 4 B, please check “CXR-TMDP”
Ans: Corrected to Cr-TMDP in the figure.
All the tables should be the three line table, keep them uniformly
Ans: We have corrected the tables accordingly.
Line 376, “regenaration” should be “regeneration”
Ans: Corrected in the text.
Line 386-393, please add figure to explain the regeneration studies, it will be clearly.
Ans: We have added the figures in the text as suggested.
References: line 416 “chemosphere” and line 461 “chemosphere”, please keep the name of the journal the same typeface.
Ans: Corrected uniformly throughout the list.
Round 2
Reviewer 2 Report
There is too short text in the paper in the "Discussion" chapter. This chapter needs to be reworked. I think it would be better if the discussion was inserted into the text in the analytical part.
Author Response
Comments and Suggestions for Authors
There is too short text in the paper in the "Discussion" chapter. This chapter needs to be reworked. I think it would be better if the discussion was inserted into the text in the analytical part.
Response: Thank you very much for suggestion. The following information has been added in the main text of the revised manuscript. There is inadequacy of research using date seed powders as adsorbents for heavy metals removal. Few researches were used date stones, raw date pits and burnt date pits as potential adsorbent rather dates pit powders for removal of heavy metals [46, 47] and they reported enhance adsorption of heavy metals on the burning of the date seeds. Mohamed et al [48] reported the Cu(II) adsorption using ajwa date pits powder and the carbonized date seeds for the removal of Cu(II) from wastewater was demonstrated by other workers [49]. Our results for Cr(III) and Cd(II) removal using MDP, SDP, TMDP and TSDP were not too different from the reported findings in terms of heavy metal adsorption, only the preparation of the adsorbents were of slight alteration. However, in the current research, both the untreated and H2O2 treated materials showed excellent adsorption efficiency towards Cr(III) and Cd(II) ions.
A series of batch experiments were investigated for optimizing the adsorbent dosage, solution pH, contact time and initial adsorbates concentrations and the metal ions were determined using atomic adsorption spectrometry (AAS). The maximum adsorption capacity for all the metal ions was noted at pH 7.8. The obtained results were found to be best fitted with Langmuir isotherm model, where the maximum adsorption capacities (monolayer) at 323 K were 1428.5 mg/g and 1302.0 mg/g (TMDP) and 1228.5 mg/g and 1182.0 mg/g (TSDP) for Cr(III) and Cd(II), respectively. The efficiencies of all the adsorbents; MDP, SDP, TMDP and TSDP, were compared and it was found that for both Cr and Cd elements, the adsorption capacities of TMDP and TSDP were much higher compared to untreated materials (MDP and SDP). However, slight higher adsorption capacity of TMDP was noticed compared to TSDP and the most efficiently removed metal ion was Cr(III) than Cd(II) for all the adsorbents. This could be attributed to the surface morphology and pore size of the adsorbents. The higher pore size and surface functionality of the H2O2 treated material compared to MDP and SDP, were indicated by FTIR, BET, and SEM results.
Reviewer 3 Report
- 3.2.5 discussion, please check.
- Fig 7, H2SO4, please use the superscript and subscript.
- %?????????? should be ??????????%
- please check all this manuscript carefully again before publication.
Author Response
Reviewer 3
Comments and Suggestions for Authors
- 2.5 discussion, please check.
Response: Thanks for sugegstion. We have corrected it in the revised manuscript.
- Fig 7, H2SO4, please use the superscript and subscript.
Response: Corrected
- %?????????? should be ??????????%
Response: Thanks for comment. We have corrected it in the revised manuscript.
- Please check all this manuscript carefully again before publication.
Response: We have thoroughly checked and corrected the manuscript accordingly.